# Mast Cell Density in Squamous Cell Carcinoma of Skin in Dogs and Cats

**DOI:** 10.3390/ani15030316

**Published:** 2025-01-23

**Authors:** Nomeda Juodžiukynienė, Kristina Lasienė, Nijolė Savickienė, Albina Aniulienė

**Affiliations:** 1Department of Veterinary Pathobiology, Faculty of Veterinary Medicine, Veterinary Academy, Lithuanian University of Health Sciences, A.Mickevičius Str. 9, LT-44307 Kaunas, Lithuania; albina.aniuliene@lsmu.lt; 2Deparment of Histology and Embryology, Faculty of Medicine, Medical Academy, Lithuanian University of Health Sciences, A.Mickevičius Str. 9, LT-44307 Kaunas, Lithuania; kristina.lasiene@lsmu.lt; 3Department of Pharmacognosy, Faculty of Pharmacy, Medical Academy, Lithuanian University of Health Sciences, A.Mickevičius Str. 9, LT-44307 Kaunas, Lithuania; nijole.savickiene@lsmu.lt

**Keywords:** mast cell, dog, cat, squamous cell carcinoma, skin tumor, Giemsa stain, pathogenesis, cancerogenesis

## Abstract

The aim of the present study was to evaluate mast cell counts in the squamous cell carcinomas of dogs and cats. Little is known about mast cells in the tumors of dogs and cats. Squamous cell carcinoma of the skin is the second most common tumor in cats after mammary carcinomas and the third most common in dogs after mastocytomas and mammary carcinomas. Mast cells play a key role in both angiogenesis and tissue remodeling (degradation of basement membrane, extracellular matrix, etc.), facilitating tumor initiation and growth. The tumorigenic effects of mast cells include angiogenesis, lymphangiogenesis, immunosuppression, disruption of the extracellular matrix and promotion of tumor cell mitosis due to the production and release of multi-potent molecules by mast cells, and the interation of these these multi-potent molecules with host stromal and immune cells. The number of mast cells was evaluated in squamous cell carcinomas of two animal species—dogs and cats. The mast cell count in Giemsa-stained histological slides was calculated. In cats, a significantly higher total mast cell count was found, as well as higher levels in intratumoral and peritumoral tissues.

## 1. Introduction

Mast cells have very wide-ranging biological characteristics [1,2] and play a central role in many processes [3,4]. Mast cell activation and mediator release have different effects in various tissues and organs [5]. They regulate vasodilation, angiogenesis, bone growth, mineral homeostasis, tissue regeneration, bacterial and parasite elimination, and the functions of many cell types, such as dendritic cells, immune cells, fibroblasts, endothelial cells, and epithelial cells [5]. Mucosal mast cells produce only tryptase and connective tissue mast cells produce chymase, tryptase, and carboxypeptidases [1,2,5,6]. Despite extensive research, mast cells remain a subject of debate and controversy due to their dual biology. Mast cells, being able to act contrary to many cells, can both stimulate and inhibit antitumor immunity [5,6]. Mast cells generate and release multi-potent molecules such as histamine, proteases, prostanoids, leukotrienes, heparin, and many cytokines, chemokines, and growth factors [7,8,9,10]. Most of them can contribute to tumor progression. The antitumoral effect of mast cells is also related to many of the aforementioned substances that can stimulate dendritic cells and T cells, act cytotoxically on tumor cells, and have an antiangiogenic action, etc. [11]. Mast cells are a part of the tumor microenvironment together with other immune cells, stromal cells, blood vessels, and the extracellular matrix [12,13,14,15].

Studies on human tumors have shown that the number of mast cells can vary markedly in different types of tumor tissues and in peritumoral tissues as well. In addition, it has been observed that, in some cases, a high number of mast cells in tumor tissues is associated with a better prognosis, while in others, it is associated with a worse one [16]. Similarly, conflicting data have been obtained about the significance of peritumoral mast cells for prognosis [13,15,16]. Skin squamous cell carcinoma is common in both humans and animals. Several conditions are necessary for a skin malignancy to develop, including a malignant transformation of skin cells and compromised immunity, especially impaired cell-mediated immune response [17]. Ultraviolet B (UVB) radiation (280–320 nm) causes tumoral transformation of epidermal cells, activation of mast cells [18] and alteration of the cell-mediated immune response, thereby enabling cutaneous tumors to escape elimination [19]. Both direct (formation of cyclobutane pyrimidine dimers (CPDs and 6-4 photoproducts) and indirect (oxygen reactive species) DNA damage can lead to mutations in genes that promote the development of skin cancers (BRAF and NRAS genes) [20]. UVB radiation suppresses the immune function of skin Langerhans cells and damages keratinocytes [21]. Skin cancers are highly immunogenic, and a competent immune system should be capable of destroying the cancer cells [19]. In humans, a suppressed immune function has been demonstrated to be a causative factor for the development of skin cancers [18]. In addition to these components, there are other components involved in the oncogenesis of SCC in humans and animals, including environment, geography, genetic factors, papillomavirus, animal fur color, and anatomical location [22,23]. Results of studies suggest that UVB indirectly activates mast cells by stimulating the isomerization of trans-uronic to urocanic acid, and this acid together with UVB suppress immune responses to sensitizing antigens. Histamine is also involved in immune response suppression [18]. White-furred cats are more prone to ear or nasal planum SCC, while black-furred, large-breed dogs are more prone to SCC of the nail beds [24].

Although skin tumors are common in animals, little is known about the role of mast cells in their development. Because the amount of mast cells in animal tumors has not been extensively studied and the data about mast cells in canine and feline tumors are sparse, the aim of this study was to determine the number of mast cells in the tissues of one of the most common tumors in dogs and cats—skin squamous cell carcinoma.

## 2. Materials and Methods

### 2.1. Ethics Statement

The specimens were received as tissue samples for routine H and E staining after tumor surgery. Tumor tissues left over from routine H and E examinations were used for research. Because the study was not based on direct contact with animals (pathological material used in routine diagnostics was used instead), approval from an animal ethics committee was not required. The animals were handled by owners according to high ethical standards and national legislation (Animal Welfare Protection and Law of the Republic of Lithuania: Žin., 1997, Nr. 108-2728; 2012, Nr. 122-6126). The owners‘ consent was obtained in the veterinary clinic to allow the use of animal tumor tissues for scientific purposes.

Origin of specimens. The skin lesions were removed at the LUHS dr. L. Kriauceliunas Small Animal Clinic following surgical recommendations. If a tumor of unknown origin was removed (without cytology or incisional biopsy before surgery), it was removed with 1 cm of healthy tissue margins in all directions. In case of mastocytoma or carcinoma, the tumour was removed with at least 2 cm of healthy tissue margins, together with the muscle fascia. Anesthesia protocol and an animal owner’s agreement were included.

### 2.2. Tissue Preparation

Specimens. Skin squamous cell carcinoma tissues from dogs (*n* = 15) and cats (*n* = 15) were examined. The archival paraffin-embedded specimens from the Pathology Center of the Veterinary Pathobiology Department of the Veterinary Academy, at the Veterinary Medicine Faculty of Lithuania University of Health Sciences were used.

Tissue proccessing. Paraffin blocks were fabricated using “Shandon Pathcentre” and “TES 99 Medite Medizintechnik” equipment. Serial 4-μm sections were prepared with a “Sakura Accu-Cut SRM” microtome from each sample and used for routine H and E and Giemsa (for mast cells) staining.

Gross morphology and anatomy. Specimens’ gross morphology was established according to specimens’ description in cover letters to the Pathology Center. According to anatomical localization, the samples were distributed as follows: in dogs, there were 10 specimens from the sides of the torso and 5 from the nail bed; in cats, there were 10 specimens from ears and 5 from the nasal planum.

### 2.3. Histotechnique

Microscopical evaluation. Histological H and E slides and Giemsa slides were evaluated and image analysis was performed using an Olympus microscope (Olympus BX41, Tokyo, Japan) equipped with a digital Olympus DP72 image camera and CellSensDimension software.

Histochemistry. Staining for mast cells (MCs) was performed using Giemsa Stain Modified Solution (Sigma-Aldrich, Egham, Surrey, UK). The slides were deparaffinized in xylene and isopropanol, rehydrated with 96% ethanol and washed in running tap water. The slides were subsequently left in Giemsa stain for 1 h. They were then submerged for 10–12 s in acidic water and 96% ethanol for colour differentiation and completion.

### 2.4. Histopathology

Histological evaluation. Covered H and E and Giemsa slides were scanned at  ×100 magnification for the evaluation of SCC grade and mitotic count (H and E) and for areas of the greatest abundance of MCs in the intratumoural and peritumoural zones (Giemsa).

The histological grade of squamous cell carcinoma in dogs and cats was established using an adapted version of Nagamine et al. [25] and Broder‘s grading system [26] at ×100–400 magnification.

The intratumoral and peritumoral zone was defined according to Glajcar et al. [27]. Mast cells located within the centre of neoplastic tissue and more than 1 HPF inside the tumor‘s edge were considered as the intratumoral population. MCs no further than 1 HPF from the tumor margins and no more than 2 HPF outside the tumor margins were considered to reside in the peritumoral zone. MCs were counted in non-overlapping HPFs of the sample. Intratumoral mast cell density (IMCD), peritumoral mast cell density (PMCD) and total mast cells density (TMCD), as a sum of IMCD and PMCD, were counted in 1 mm^2^ (FN 22/40× objective = 0.55 diameter; area = πr^2^ = 0.237 mm^2^ × 10 fields = 2.37 mm^2^).

The connective tissue, both around the tumor and inside it (tumor stroma), had a dual appearance—it was found to be both loose and well-vascularized, and dense, compact and fibrous. For this reason, the mast cells in canine and feline squamous cell carcinomas were counted separately in all four categories.

### 2.5. Statistical Analysis

Statistical analyses were performed using the SPSS 20 (IBM, Chicago, IL, USA) statistical software package. Data were expressed as mean ± standard deviation. A one-way ANOVA test was used to determine differences between groups. Differences at the value of *p* < 0.05 were considered significant.

## 3. Results

The anatomical distribution of the samples was as follows: in dogs, samples were from the trunk of the body (*n* = 10) and from the nail bed (*n* = 5); in cats, samples were from the ears (*n* = 10) and the nasal planum (*n* = 5). According to the gross description, skin squamous cell carcinomas (SCC) in dogs and cats were very different in appearance and the size of skin lesions (nodules, ulcers, abscess-like formations, crater-like lesions), see Figure 1 and Figure 2.

Histological grades of SCC varied from 1 to 2, nuclear pleomorphism was moderate, and the infiltration pattern varied from cords, bands and strands to small islands or cords of cells. Complete invasion of the dermis, outward growth of tumor masses into subcutaneous tissues, destruction of cartilage (ears) and bone destruction (fingernail bed) were observed in routine H and E slides, see Figure 3 and Figure 4. Between two and five mitoses per HPF were observed.

In all samples, the connective tissue, both around the tumor and inside it (tumor stroma) had a dual appearance—it was found to be both loose and well-vascularized and dense, compact, and fibrous. The loose, well-vascularized tissue contained abundant immune cells, and the dense fibrous tissue contained a small number of immune cells.

For this reason, we counted mast cells separately in areas of loose and dense fibrous tissue both inside and around the tumors (four tissue categories). Mast cell distribution in all four tissue categories differed between canine and feline SCC and is present in Table 1 and Figure 5, Figure 6, Figure 7, Figure 8, Figure 9 and Figure 10.

Loose, well-vascularized connective tissue in both dogs and cats was found to be variably abundantly infiltrated with inflammatory cells. This tissue was found to be richer in neutrophils in some cases, and richer in lymphocytes and plasmocytes in others. It was observed that mast cells were found more abundantly in lymphocyte- and plasmocyte-rich tissues than in neutrophil-rich tissues (in general, areas with neutrophils were devoid of mast cells or had single cells). Areas rich in lymphocytes and plasmocytes had a different number of mastocytes. Lymphocyte- and plasmocyte- rich areas had varying amounts of mast cells—abundant mast cells were seen in some areas, and single ones were seen in others. Likewise, fibrous tissue, which had an overall lower number of inflammatory cells, had more mast cells in some cases and fewer in others.

A very marked difference in the number of mast cells in the SCC tissues was found between animal species.

The following tendency was found in both dogs and cats: a significantly higher number of mastocytes was found in both peritumoral and intratumoral loose connective, well-vascularized tissue. Conversely, lower numbers of mast cells were found in both intratumoral and peritumoral compact fibrous tissue in both animal species.

In dogs, a significantly higher number of mast cells in peritumoral loose, well-vascularized tissue was observed compared to the other three categories (*p* < 0.05). However, among themselves, these three categories differed significantly in the number of mast cells (Figure 11).

A different expression of mast cell quantity in different tissue categories was observed in cats. Significantly more mast cells were found in both peritumoral and intratumoral loose, well-vascularized tissues (188.5 ± 100.03 and 183.27 ± 92.23 respectively) compared to fibrous peritumoral and intratumoral tissues (14.46 ± 8.3 and 16.8 ± 6.49, respectively, *p* < 0.05). However, there were no significant differences between mast cell counts in peritumoral and intratumoral loose, well-vascularized tissues. A significantly lower number of mast cells was also found in peritumoral and intratumoral fibrous tissue (14.46 ± 8.3 and 16.8 ± 6.49, respectively, *p* > 0.05) (Figure 12). There were no significant differences in mast cell counts between intratumoral and peritumoral fibrous tissue regions.

Thus, in felines, mast cell content in both peritumor and intratumoral loose, well-vascularized tissue differed markedly from that in fibrous peritumoral and intratumoral tissue (*p* < 0.05).

The results of this study revealed marked differences in the number of mast cells in canine and feline SCC. In cats, a significantly higher number of mast cells was found in both intratumoral and peritumoral loose, well-vascularized tissues.

Perhaps the markedly higher number of mast cells in cats was due to the predominant ear carcinoma tissue samples, in which a very high number of mast cells was found in the loose, well-vascularized connective tissue.

## 4. Discussion

We found a significant difference in the number of mast cells in 1 mm^2^ SCC tissues of dogs and cats—211.83 ± 46.15 and 403.01 ± 107.34, respectively. The high mast cell count in carcinoma tissues in cats, we believe, is related to the predominance of ear samples, in which very high mast cell counts were found. SCC of the ears, when already developed, continues to be exposed to the sun and progresses if left untreated, with sun exposure contributing to sun-induced inflammation in the tissues of the ears. It is also unclear whether the animals were symptomatically treated with NSAIDs or glucocorticoids, either systemically or topically, prior to sampling, and to what extent the animals themselves could have accessed the affected areas (i.e., the areas where the tumour was developing) by scratching with their teeth (as such areas are itchy and painful) and thereby mechanically irritating and infecting the ulcerated tumor areas, which may have complicated the inflammatory process further.

A recent study has shown an increase in mast cells (MCs) in chronically sun-exposed human skin [28]. Chronic sun exposure has been described as a significant risk factor. However, data on the influence of the sun on animal carcinoma are not yet reliable [29]. Certain breeds and coat colors in dogs are a risk factor [30]. In humans, it has been suggested that UV light and human beta papillomaviruses may act together to cause cutaneous SCC [31,32]. Papillomavirus (FcaPV6) was identified in the squamous cell carcinoma of the nasal planum in domestic cats [33]. Canine oral papillomavirus has not been shown to cause squamous cell carcinomas (SCC) [32] or nail-bed SCC in dogs [30]. In the analyzed histological samples, a marked variation in the composition of inflammatory cells was detected. In tumor tissue with abundant neutrophil infiltrates, we found very few or no mast cells, whereas in tissue with lymphocytes, they were present in varying amounts. Sources from the literature confirm that feline SCC tissues may contain lymphoplasmocytic, neutrophilic or mixed infiltration, and in other cases, a prevalence of mast cells can be observed [29]. Chemokines and cytokines released by MCs can change the initial function of B- and T-lymphocytes, and of other immune system cells, and this immunomodulation might either enhance or suppress tumor growth [34]. The presence of inflammatory cells in peritumoral and intratumural tissues is considered a host reaction to neoplastic tissue and is found in many specimens of SCC [35]. Both intratumoral and peritumoral fibrous tissues were characterized by a significantly lower number of mast cells in dogs as well as in cats, while well-vascularized tissues of both peritumoral and intratumoral types in both species had more mast cells. We identified fibrous connective tissue as the area of the tissue that consisted of abundant, relatively thick fibers and fibroblasts oriented in one direction, with directionally arranged blood vessels. More abundant fibrous tissue was observed in canine SCC, particularly in the nail bed. Slayter and Boosing (1994) found that there is often a desmoplastic reaction in the surrounding dermis and subcutaneous tissue in response to the invasive neoplastic squamous epithelial cells [36]. Extensive fibroblast proliferation of the dermis was often observed in response to the infiltrating neoplastic cords of SCC in reptiles [37]. Lung SCC with a fibrous stroma is more aggressive, and this suggests that tumor cells and peritumoral fibroblasts can create a tumor-favorable microenvironment [38]. The connective tissue with its own cells and immune cells forms a microenvironment and is closely related to the invasive front of the tumor [25]. Neutrophils, like macrophages, are highly abundant immune cells in a large number of cancer tissues [28] and can have both pro- or anti-tumor properties [39]. Neutrophils promote pancreatic tumor angiogenesis mainly by secreting MMP-9 [40,41]. Extensive infiltration of SCC connective tissue by neutrophils is associated with more aggressive SCC [35].

Mast cells contribute to the development of many tumors: squamous cell carcinoma of the skin and lips, non-small lung carcinoma [42], basal cell carcinoma, melanoma [43] and others. The effect of mast cells on tumor progression is highly controversial [8,42].

High intratumoral MC density was associated with a worse prognosis in the pancreatic duct adenocarcinoma [16,44] and with a good prognosis in prostate cancer [45]. High concentrations of peritumoral mast cells were associated with a poor prognosis in prostate cancer [45]. However, a low density of peritumoral mast cells was associated with a worse prognosis in stage I non-small cell lung adenocarcinoma [42].

Another study shows associations with a good prognosis [46]. No correlation was found between MCs and the prognosis in invasive breast carcinoma [47]. In some solid tumors, MCs are detected in the intratumoral areas, while in others, they are preferentially located in the peritumoral zones. Interestingly enough, the presence of peritumoral MCs seems to indicate a bad prognosis, while its intratumoral location is associated with both favorable and unfavorable prognoses [47]. This discordance can be explained in part by the high heterogeneity of MCs biology [11].

## 5. Conclusions

A major drawback of this study was the small number of histological samples, which may have led to a marked scatter in the data. Furthermore, the study samples were markedly heterogeneous in their anatomical location, which determines both the biology of SCC and the influence of different external factors.

We found a significant difference in the number of mast cells in 1 mm^2^ SCC peritumoral and intratumoral tissues in dogs and cats. A high number of mast cells in cats is believed to be associated with both the tumor process and the inflammatory process (in the case of ear carcinoma, solar radiation contributes to the inflammatory reaction). Cats have more intense and aggressive inflammatory processes than other animals (e.g., persistent lymphocytic plasmacytic inflammations, eosinophilic inflammations), and cats‘ fibroblasts are more susceptible to damage (e.g., injection-site sarcoma), which suggests that they possibly have a peculiar immune response. Mastocytemia and visceral metastases are common in feline mastocytomas.

The significantly higher number of mast cells in feline carcinomas suggests that mast cells are more intensively involved in the oncogenesis of these animals, and possibly also in the inflammation process, than in dogs

This study proves that there are species differences in both inflammation and the tumor process. Therefore, detailed studies are needed to investigate the microenvironment of animal tumors and its interaction with tumor cells.

## Figures and Tables

**Figure 1 animals-15-00316-f001:**
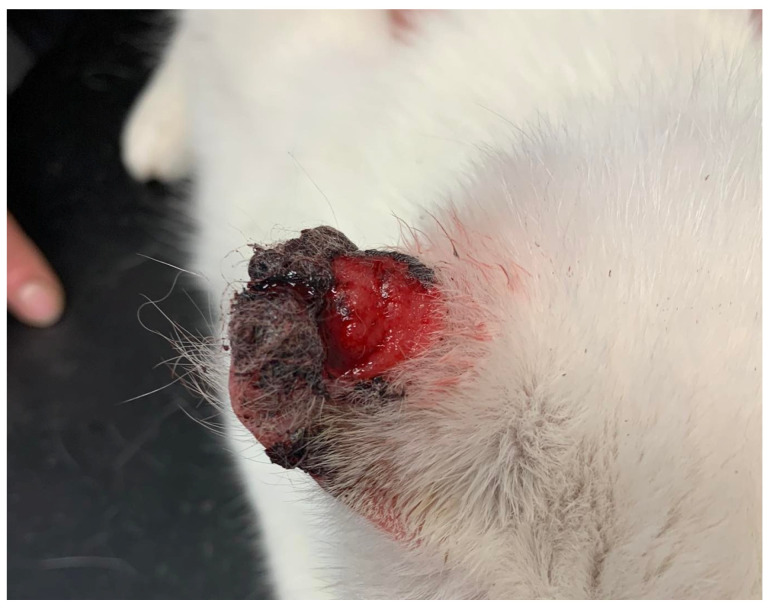
Squamous cell carcinoma affecting half auricle in cats.

**Figure 2 animals-15-00316-f002:**
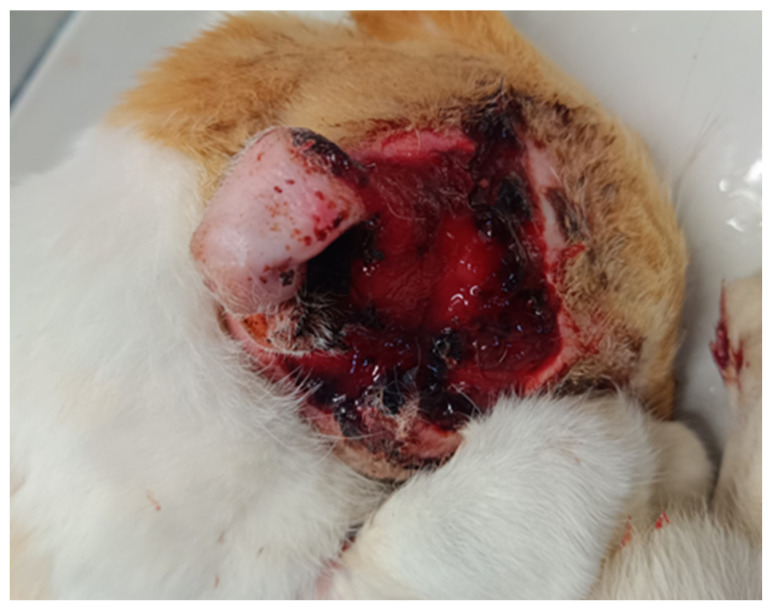
Squamous cell carcinoma that has spread widely in the surrounding tissue region.

**Figure 3 animals-15-00316-f003:**
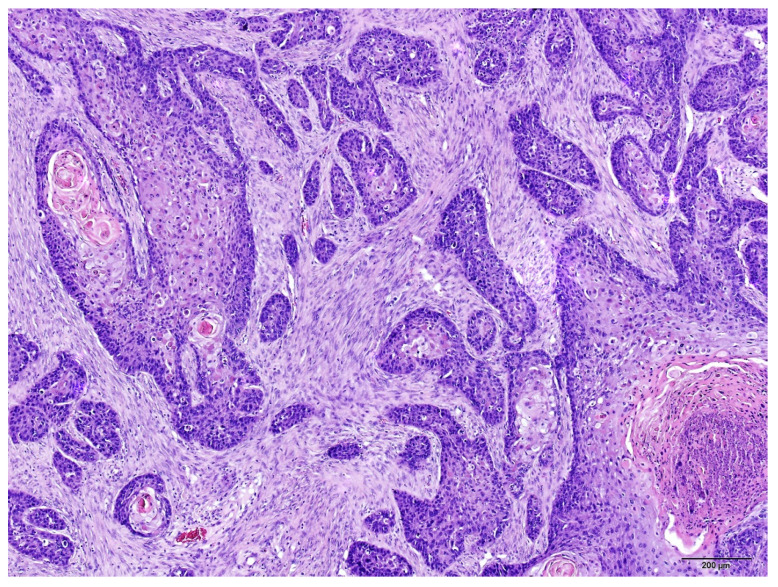
Feline ear skin SCC, grade 1. Tumor cells form large nests with typical squamous differentiation and keratinization and small nests with slight keratinization. Fibrous stroma with sparse lymphocytes is visible. H and E, ×100 magnification.

**Figure 4 animals-15-00316-f004:**
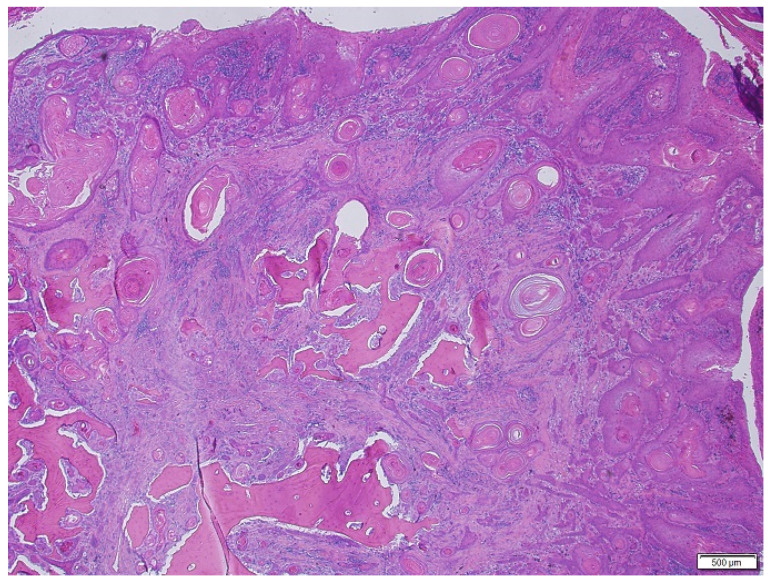
Canine nail bed SCC, grade 1. Tumor cells form large nests with typical squamous differentiation and keratinization and small nests with slight keratinization. Stroma is moderately–severely infiltrated with a mixed inflammatory cell population. Bone destruction is visible. H and E, ×40 magnification.

**Figure 5 animals-15-00316-f005:**
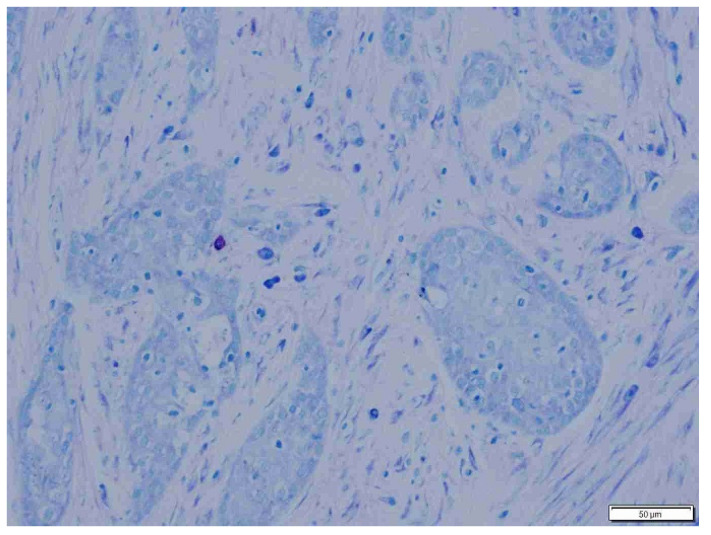
Sparse mast cells in fibrous peritumoral connective tissue in dogs. Giemsa, ×200 magnification.

**Figure 6 animals-15-00316-f006:**
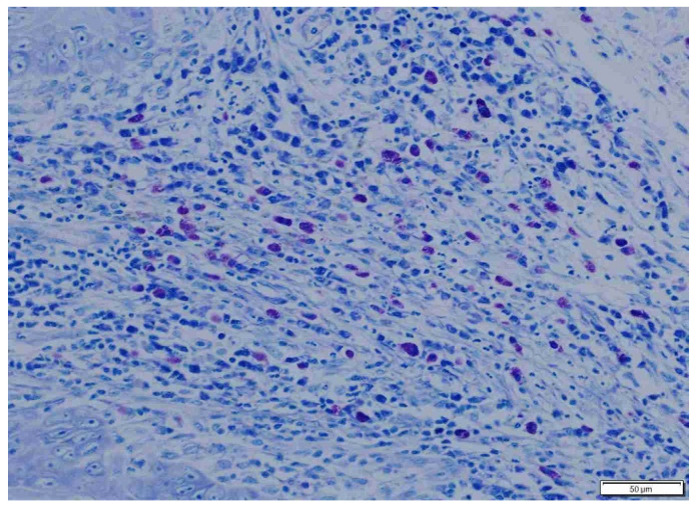
Abundant mast cells in loose, well-vascularized peritumoral connective tissue in cats. Giemsa, ×200 magnification.

**Figure 7 animals-15-00316-f007:**
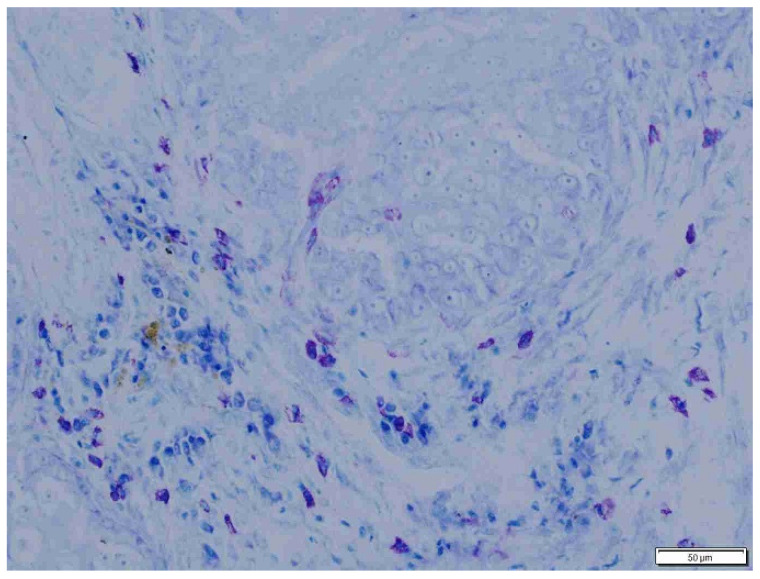
Moderately abundant mast cells in fibrous intratumoral connective tissue in dogs. Giemsa, ×200 magnification.

**Figure 8 animals-15-00316-f008:**
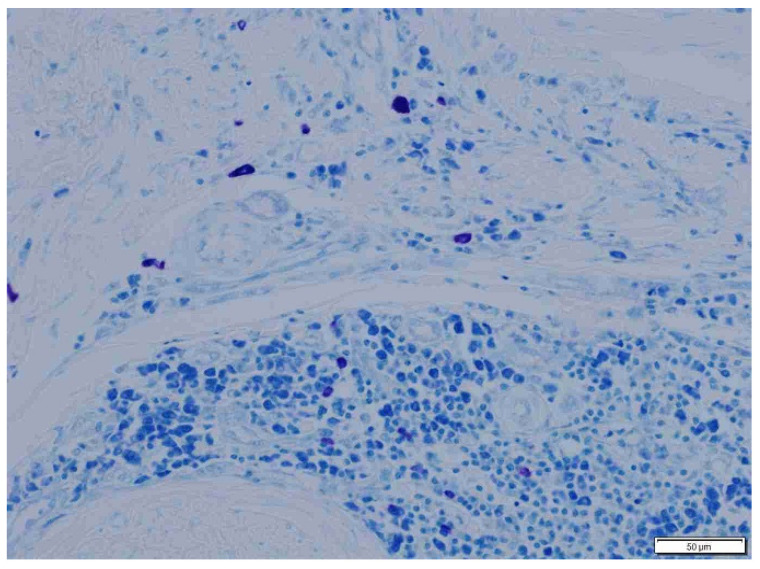
Sparse mast cells in loose, well-vascularized inratumoral connective tissue in cats. Giemsa, ×200 magnification.

**Figure 9 animals-15-00316-f009:**
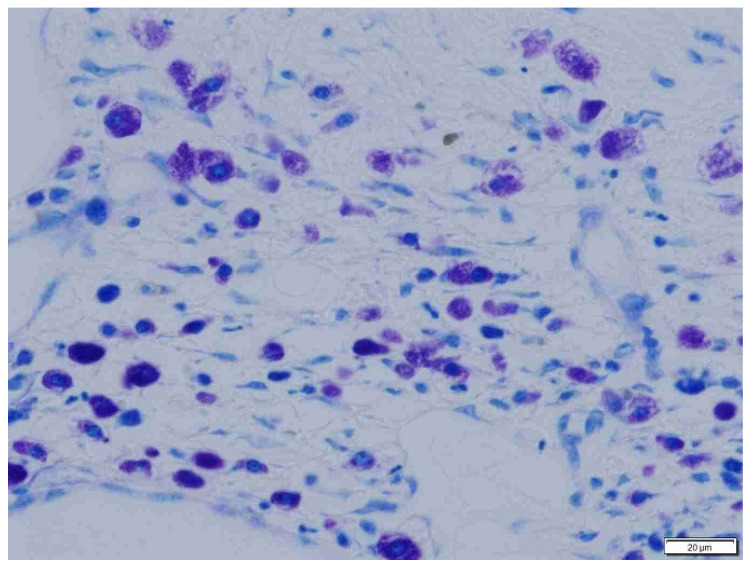
Abundant mast cells in loose, well-vascularized intratumoral connective tissue in cats. Giemsa, ×400 magnification.

**Figure 10 animals-15-00316-f010:**
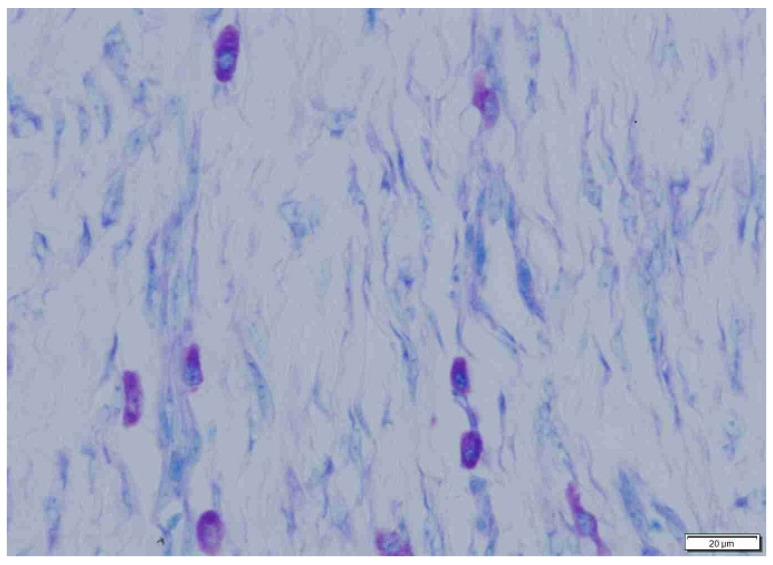
Moderately abundant mast cells in fibrous intratumoral connective tissue in dogs. Giemsa, ×400 magnification.

**Figure 11 animals-15-00316-f011:**
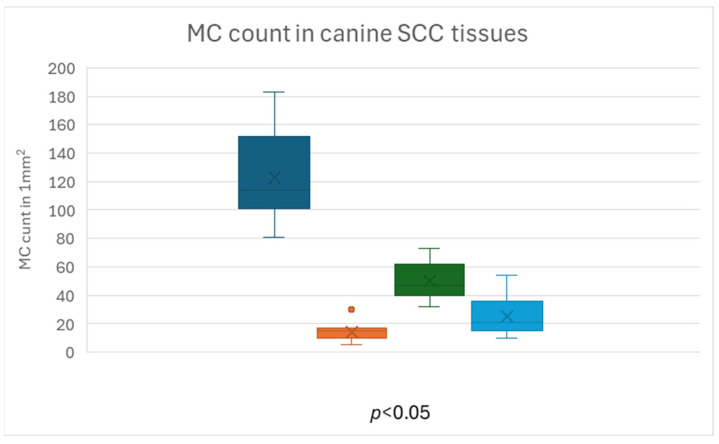
Mast cell distribution in canine SCC tissues: dark blue—PTMC in loose tissue, orange—PTMC in fibrous tissue, green—ITMC in loose tissue, light blue—ITMC in fibrous tissue.

**Figure 12 animals-15-00316-f012:**
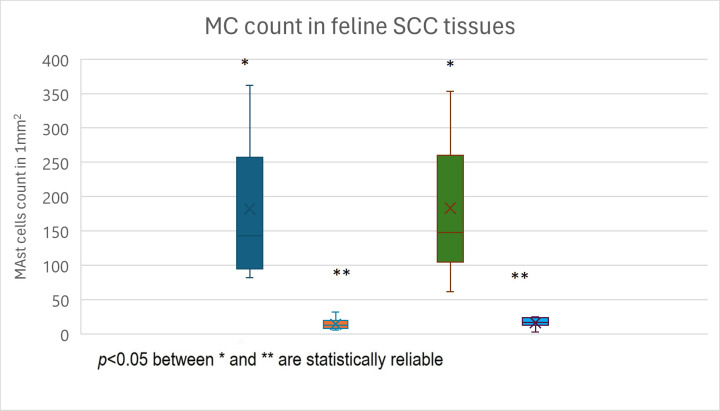
Mast cell distribution in feline SCC tissues: dark blue—PTMC in loose tissue, orange—PTMC in fibrous tissue, green—ITMC in loose tissue, light blue—ITMC in fibrous tissue.

**Table 1 animals-15-00316-t001:** Mast cell count in peritumoral and intratumoral tissues in canine and feline squamous cell carcinomas (in 1 mm^2^; mean ± standart deviation).

	Canine SCC	Feline SCC
**1.** **Peritumoral loose connective tissue**	122.6 ± 70.57 a1	188.5 ± 100.03 b1
**2.** **Peritumoral fibrous connective tissue**	13.93 ± 13.93 a2	14.46 ± 8.31 b2
**Total PTMC**	**136.53** ± 59.34 a3	**202.96 ± 110.21 b3**
**3.** **Intratumoral loose connective tissue**	50.3 ± 12.88 a4	183.27 ± 92.23 b4
**4.** **Intratumoral fibrous connective tissue**	25.0 ± 12.95 a5	16.8 ± 6.49 b5
**Total ITMC**	**75.3** ± 18.06 a6	**200.1** ± 106.27 b6
**Total MC**	**211.83** ± 46.15 **a7**	**403.01** ± 107.34 b7

a1:a2; a4:a5; b1:b2; b4:b5, *p* < 0.05. a1:b1; a3:b3; a4:b4; a5:b5; a6:b6; a7:b7, *p* < 0.05.

## Data Availability

The data presented in this study are included within the article. The raw data supporting the findings of this study are available from the corresponding author upon reasonable request.

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
