# Peer review of "Mast Cell Density in Squamous Cell Carcinoma of Skin in Dogs and Cats"

_animals, 2025, doi:10.3390/ani15030316_

Round 1
Reviewer 1 Report
Comments and Suggestions for Authors
The Abstract is quite short. A description of the purpose of the research is definitely missing. It must be added.
The material and methods are also short. The description of material is lacking (the most important part - location, details about keratinization or mitotic count?). I also suggest to add the description of how mast cells were counted (peritumoral and intratumoral, how many foci).
In results you mention about four categories, but there is no explanation of what they are.
What does it mean "medium magnification"?
What I feel is definitely missing from the discussion is more reference to the presence of mast cells in the tumour environment and their impact on tumour growth. I also agree with the statement that a weak point is the small number of samples studied - especially as the authors themselves mention that this is a common tumour.
The references should be revised in accordance with the journal's guidelines.
Comments on the Quality of English LanguageThere is a mix of British and American English. There are also spelling errors (eg wel, should be well). Please also change fibrotic into fibrous.
Author Response
Dear Rewiever,
We agree with the comments, especially that we did not notice that the part of „Methods of description was mising from the article when the text was uploaded.
- The Abstract is quite short. A description of the purpose of the research is definitely
missing. It must be added.
We fixed this problem.
The material and methods are also short. The description of material is lacking (the most
important part - location, details about keratinization or mitotic count?). I also suggest to
add the description of how mast cells were counted (peritumoral and intratumoral, how
many foci).
We fixed this problem.
The part „Methods“ was missed from article before uploading...
- In results you mention about four categories, but there is no explanation of what they are.
Exlanation is in the part „Methods“.
- What does it mean "medium magnification"?
We fixed this probem. We meant a 400x magnification here.
- What I feel is definitely missing from the discussion is more reference to the presence of
mast cells in the tumour environment and their impact on tumour growth.
We fixed this problem.
- I also agree with the statement that a weak point is the small number of samples studied - especially as the authors themselves mention that this is a common tumour.
Yes, we agree. we will avoid this kind of mistake in the future
- The references should be revised in accordance with the journal's guidelines
We fixed this problem.
Sincerely, authors

Reviewer 2 Report
Comments and Suggestions for Authors
For My dear/Author
II-Abstract :
There are no highlights
There are no graphical abstract
What is the relationship between the mast cells and SCC
What about the bird or cell nests ?
How we can differentiate the SCC from mastocytoma ?
What do you mean by mast cells density--number /infiltration or its relation to the other skin cells ?
LN/14-18--the simple summary --can considered as a high lights or background
LN/16--tissue remodeling---explain and usually used for bone (remodelling )
LN/17---detailed pathogenesis should be applied
Mast cell initiator to SCC--again in chart describe the molecular pathways and the genetic abnormalities
LN/19-28---abstract is very short and should contain the following items: Background-aims/methods/results and conclusion
LN/22---what do you mean by died dark purple ?
What about the clinical sings noticed on dogs and cats before dislocation ?
A copy of necropsy report should be applied
LN/22-24--measured by what ?
LN/26---neutrophil abundant with the tumor---your clarification ?
LN/29--add histochemical stains/patho-pets ,skin tumors to the keywords
Introduction :
LN/32---explain why ??? and add reference
LN/40-41---pathogenesis ???
LN/42--TME--detailed then abbreviate
LN/44---delete a double edged sword
LN/45---add reference---protect against some tumors---describe this again
LN/46--pro-tumorigenesis function---molecular pathway in chart ???
LN/50---add reference
LN/53---double effects---not sufficient
Huge number of abbreviations were used---tabulate all
Introduction is very long -- be more concise
Aims of the present study are not present at the end of the introduction
Novelty of this study should be more highlighted and more adjusted
Materials and methods :
The most descriptive methodologies are without references--why ?
There are no plan for the study area ?
LN/72--write as H&E stain
LN/91---write as histological analysis
Complete details about the paraffin embedding technique should be applied and staining methods either for H&E or Giemsa should be associated with an updated references
A copy of necropsy list should be enclosed
Tabulate all gross abnormalities before the surgical excision
Results
LN/112---what is/are the cause of the dual appearance(mixed) stromal tumor ?
Figure: 1,2,3,4--all stained with Giemsa not H&E as a routine stain and why you did not use tumor markers for more confirmation ??? also what about the magnification power recorded--H&E--X----etc----
The authors did not mention any thing about :-
Gross figures of the tumors before excision (shape/color/consistency/C/s--etc)
H&E stained figures---mentioned at the intro
The clinical signs of the investigated dogs/cats before exposure to surgery
Which type of surgery used and anaewsthesia ???
The percentages of mortalities --if there
Write as Table ( ):---------------/Fig.( ):----------etc--apply for all
Results should be improved and additional gross figure and H&E stained slides applied
Discussion
LN/178--explain why the abundance of neutrophil ? and what about the well-known message that increased the angiogenesis and the leukocytes used by the tumor for helping in metastasis not for host protection ????
LN/195--two different style of writing the references were detected--why ??? same style should be
Discussion is very long and should be based upon debating the obtained results with those of the previous investigators results
Conclusion :
Less than enough
References :
Some cited references need to be more update
Delete pp from all
We don't add etal with the references(Ref,4---etc) , unless the total number of authors exceed than 6 , hence , add etal with the last ones--apply for all
Huge number of references were used (55)???
Some journal names were written abbreviated , while others were not--why ?? same style should be
Author Response
Dear rewiever,
We thank for helpful comments. We have improved the article according to your comments.
I am to blame for the many flaws in the article.
I wrote article in a hurry because I really didn't have time to submit the article...
Your comments were helpful, but there are a few things I want to clarify.
I present your comments in order and to each of them - our answer
There are no highlights
We fixed the problem.
There are no graphical abstract
We fixed the problem.
What is the relationship between the mast cells and SCC
We fixed the problem in introduction.
What about the bird or cell nests ?
Yes, we did not describe them in detail. we fixed the problem
How we can differentiate the SCC from mastocytoma ?
SCC and mastocytoma are tumors of different etiology, biology and histology. Both its macro and micro image are different. The simple rutine HE microscopy shows specific histo images for both carcinoma (depending on the degree of differentiation - Broder or other systems) and mastocytoma (skin grade I, II, III, subcutaneous).
What do you mean by mast cells density--number /infiltration or its relation to the other skin cells ?
The term density is used in various studies and articles to refer to the amount of mast cells (or other test objects such as blood capillaries) per 1mm2 or 2.37mm2 or 10 HPF.
LN/14-18--the simple summary --can considered as a high lights or background
We took note of the comment .
LN/16--tissue remodeling---explain and usually used for bone (remodelling )
The term remodeling is also used to refer to tumors (extracellular material, basement membrane degradation, and et.c.). sometimes used in literature to refer to regeneration. thus, the term is used to refer to more than just bone remodeling
LN/17---detailed pathogenesis should be applied
We tried to fix problem and have uploaded a short pathogenesis, because a detailed description would make the article much larger
Mast cell initiator to SCC--again in chart describe the molecular pathways and the genetic abnormalities
We tried to fix problem and have uploaded a short pathogenesis, because a detailed description would make the article much larger
LN/19-28---abstract is very short and should contain the following items: Background-aims/methods/results and conclusion
we fixed the problem
LN/22---what do you mean by died dark purple ?
Dyed....
What about the clinical sings noticed on dogs and cats before dislocation ?
We are not clinicians, except for me (N.J. -in addition to teaching, I also work as a diagnostician in a clinic). Therefore, we do not see many patients and cannot tell the clinic. We systematized and concisely presented what information was provided about clinical features in the „Methods“. Likewise, the accompanying slips for coiled specimens arriving at the pathology center did not always contain detailed information about the lesion.
A copy of necropsy report should be applied
No study animals were necropsied. As mentioned, these were histological archival samples. Samples obtained during surgery for tumor removal and routine testing.
LN/22-24--measured by what ?
We fixed the problem.
Yes, there wasn't a whole paragraph in research methods because it got deleted by accident... the whole paragraph is already loaded. and upload it to you.
2.4. Histopathology
Histological evalution. Covered HE and Giemsa slides were scanned at ×100 magnifica-tion for evaluation SCC grade and mitotitc count (HE) and for areas of the greatest abun-dance of MCs in the intratumoural and peritumoural zones (Giemsa).
Histological grade of squamous cell carcinoma in dogs and cats was established using adapted of Nagamine et al. grading system [25] at x100-400 magnification.
The intratumoural and peritumoral zone was defined according to Glajcar et al. [26]. Mast cells located within the centre of neoplastic tissue and more than 1 HPF inside the tumor edge were considered as the intratumoral population. While MCs no further than 1 HPF from the tumor margins and no more than 2 HPF outside tumor margins were considered to reside in peritumoral zone. MCs were counted in non-overlapping HPFs of sample. In-tratumoral mast cell density (IMCD), peritumoral mast cell density (PMCD) and total mast cells density (TMCD) as a sum of IMCD and PMCD were counted.
The connective tissue, both around the tumor and inside it (tumor stroma), had a dual appearance – it was found to be both loose, well-vascularized, and dense, compact, fi-brous. For this reason, the mast cells in dogs and cats squamous cells carcinomas were counted separately in all four categories.
LN/26---neutrophil abundant with the tumor---your clarification ?
Since the composition of the cells is mixed, it can also be associated with the heterogeneity of the invasive front of the tumor and additional inflammation - if it is ear samples - due to sun exposure, if other localization (nail bed, sides) - mechanical irritation with the animal's teeth, because the animal scratches such places, because they are usually it hurts, stings.
A higher number of neutrophils, at least in the samples we studied, was not significantly associated with the aggressiveness of the invasive front.
LN/29--add histochemical stains/patho-pets ,skin tumors to the keywords
Introduction :
We fixed the problem
LN/32---explain why ??? and add reference
We fixed the problem
LN/40-41---pathogenesis ???
We tried to fix problem and have uploaded a short pathogenesis, because a detailed description would make the article much larger
LN/42--TME--detailed then abbreviate
We fixed the problem
LN/44---delete a double edged sword
We fixed the problem
LN/45---add reference---protect against some tumors---describe this again
We fixed the problem
LN/46--pro-tumorigenesis function---molecular pathway in chart ???
We tried to fix problem and have uploaded a short pathogenesis, because a detailed description would make the article much larger.
LN/50---add reference
We fixed the problem
LN/53---double effects---not sufficient
We fixed problem.
Huge number of abbreviations were used---tabulate all
in our opinion, not many abbreviations were used, only to indicate squamous cell carcinoma, the total amount of mast cells, peritumoral and intratumoral amount of mast cells.
Introduction is very long -- be more concise
We tried to shorten it, but not very successfully...
Aims of the present study are not present at the end of the introduction
We fixed the problem
Novelty of this study should be more highlighted and more adjusted
We fixed the problem
Materials and methods :
The most descriptive methodologies are without references--why ?
For the histological grading and mast cell counting methodologies we have added reference.
Is it appropriate to look for a reference for a macroscopic description of the tumor?
The part of the methods section on the determination of intratumoral, peritumoral fields, determination of histological grade of the tumor was accidentally deleted when uploading the article. This section has now been added.
2.2. Tissue preparation
Specimens. The skin squamous cells carcinoma tissues from dogs (n=15) and cats (n=15) were examinated. The archival paraffin embedded specimens of Veterinary Pathobiology Department Pathology center (of Veterinary Academy, Veterinary Medicine Faculty, Lith-uania University of Health Sciences) were used.
Tissues proccessing. The paraffin blocks were made using „Shandon Pathcentre“ and „TES 99 Medite Medizintechnik“ equipment. Serial 4-μm sections were prepared with a “Sakura Accu-Cut SRM“ microtome from each sample and served for routine H&E and Giemsa (for mast cells) staining.
Gross morphology and anatomy. Specimens gross morphology (accordingly to specimens description in cover letters to Pathology Center). According to anatomical localization, the samples were distributed as follows in dogs and cats: in dogs were 10 specimens of the sides of the torso and 5 specimens of nail bed; in cats were 10 of ears and 5 specimens of nasal planum. According gross description, dogs and cats skin squamous cells carci-nomas (SCCas) were very different appearance skin lessions (nodule, ulcers, abscess like formation, crater like lessions).
2.3. Histotechnique
Microscopical evaluation. Histological H&E slides and Giemsa slides were evaluated and image analysis was performed using the Olympus microscope (Olympus BX41Tokyo, Ja-pan) supplied with a digital Olympus DP72 image camera with CellSensDimension software.
Histochemistry. Staining for mast cells (MCs) was performed with Giemsa Stain Modified Solution (Sigma-Aldrich, Egham, Surrey, UK). The slides were deparaffinised in xylene and isopropanol, rehydrated with 96% ethanol and washed in running tap water. The slides were subsequently left in Giemsa stain for 1 h. They were then submerged for 10–12 s in acidic water and 96% ethanol for colour differentiation and completion.
2.4. Histopathology
Histological evalution. Covered H&E and Giemsa slides were scanned at ×100 magnifica-tion for evaluation SCC grade and mitotitc count (H&E) and for areas of the greatest abundance of MCs in the intratumoural and peritumoural zones (Giemsa).
Histological grade of squamous cell carcinoma in dogs and cats was established using adapted of Nagamine et al. grading system [28] at x100-400 magnification.
The intratumoural and peritumoral zone was defined according to Glajcar et al. [29]. Mast cells located within the centre of neoplastic tissue and more than 1 HPF inside the tumor edge were considered as the intratumoral population. While MCs no further than 1 HPF from the tumor margins and no more than 2 HPF outside tumor margins were considered to reside in peritumoral zone. MCs were counted in non-overlapping HPFs of sample. In-tratumoral mast cell density (IMCD), peritumoral mast cell density (PMCD) and total mast cells density (TMCD) as a sum of IMCD and PMCD were counted.
The connective tissue, both around the tumor and inside it (tumor stroma), had a dual appearance – it was found to be both loose, well-vascularized, and dense, compact, fi-brous. For this reason, the mast cells in dogs and cats squamous cells carcinomas were counted separately in all four categories.
There are no plan for the study area ?
What should it look like?
LN/72--write as H&E stain
We fixed this problem.
LN/91---write as histological analysis
The part of the methods section on the determination of intratumoral, peritumoral fields, determination of histological grade of the tumor was accidentally deleted when uploading the article. This section has now been added (see above).
Complete details about the paraffin embedding technique should be applied and staining methods either for H&E or Giemsa should be associated with an updated references
Almost all articles do not cite HE and Giemsa sources, as indeed each scientific institution often uses its own improved method.
H&E has been used in our pathology center for a very long time and we don't even really know where it comes from.
A copy of necropsy list should be enclosed
Since the work is on the topic of histopathology, the material was collected from the pathology archive. All the animals that had their tumors were removed survived, or at least we don't know how they ended up. That means we did not perform any autopsy.
Tabulate all gross abnormalities before the surgical excision
Since the work is on the topic of histopathology, the material was collected from the pathology archive.
Results
LN/112---what is/are the cause of the dual appearance(mixed) stromal tumor ?
This result area does not refer to a double tumor, (did i misunderstand your question?but to a different connective tissue around the tumor and inside the tumor: loose well-vascularized and fibrous.
Figure: 1,2,3,4--all stained with Giemsa not H&E as a routine stain and why you did not use tumor markers for more confirmation ??? also what about the magnification power recorded--H&E--X----etc----
routine HE was performed (this is also mentioned in the methods), and the degree of tumor differentiation (keratinization, mitoses, invasive front, etc.) was assessed during it. IHC was not required to confirm squamous cell carcinoma because of its highly characteristic HE picture
The authors did not mention any thing about :-
Gross figures of the tumors before excision (shape/color/consistency/C/s--etc)
I uploaded two photos from my collection. photos taken during the autopsy with students. these are not the same animals whose samples were analyzed (because the corpses were old, not suitable for histological examination).
We certainly did not have the opportunity to be near the operated animals or to participate in the examination and photograph the tumors.
H&E stained figures---mentioned at the intro
We fixed problem
The clinical signs of the investigated dogs/cats before exposure to surgery
The paraffin embedded specimens were used in our study....
Which type of surgery used and anaewsthesia ???
We did not perform the operations ourselves. But I am attaching the anesthesia protocol. And I added the principle of surgical resection to the research methods
The percentages of mortalities --if there
The paraffin embedded specimens were used in our study....
Write as Table ( ):---------------/Fig.( ):----------etc--apply for all
We fixed the problem.
Results should be improved and additional gross figure and H&E stained slides applied
We fixed the problem.
Discussion
LN/178--explain why the abundance of neutrophil ? and what about the well-known message that increased the angiogenesis and the leukocytes used by the tumor for helping in metastasis not for host protection ????
We fixed the problem.
LN/195--two different style of writing the references were detected--why ??? same style should be
We fixed the problem.
Discussion is very long and should be based upon debating the obtained results with those of the previous investigators results
We fixed the problem.
It was not possible to shorten it very well
Conclusion : Less than enough
We fixed the problem.
References :
Some cited references need to be more update
We fixed the problem.
Delete pp from all
We fixed the problem.
We don't add etal with the references(Ref,4---etc) , unless the total number of authors exceed than 6 , hence , add etal with the last ones--apply for all
We fixed the problem.
Huge number of references were used (55)???
.......
Some journal names were written abbreviated , while others were not--why ?? same style should be
We fixed the problem.
The owners agreement example. anestheasia protocol, example of cover letter are added to general folder (zipped)

Round 2
Reviewer 1 Report
Comments and Suggestions for Authors
I have no comments. The manuscript is improved.
Reviewer 2 Report
Comments and Suggestions for Authors
Authors have made changes as per comments given , no further suggestions
Comments on the Quality of English LanguageIt is okay